# On the Choice of Interpolation Scheme for Neural CDEs

**James Morrill**  *james.morrill.6@gmail.com*
*Department of Mathematics*
*University of Oxford*

**Patrick Kidger**  *kidger@maths.ox.ac.uk*
*Department of Mathematics*
*University of Oxford*

**Lingyi Yang**  *yangl@maths.ox.ac.uk*
*Department of Mathematics*
*University of Oxford*

**Terry Lyons**  *tlyons@maths.ox.ac.uk*
*Department of Mathematics*
*University of Oxford*

**Reviewed on OpenReview:** *https://openreview.net/forum?id=caRBFhxXIG*

## Abstract

Neural controlled differential equations (Neural CDEs) are a continuous-time extension of recurrent neural networks (RNNs), achieving state-of-the-art (SOTA) performance at modelling functions of irregular time series. In order to interpret discrete data in continuous time, current implementations rely on non-causal interpolations of the data. This is fine when the whole time series is observed in advance, but means that Neural CDEs are not suitable for use in *online prediction tasks*, where predictions need to be made in real-time: a major use case for recurrent networks. Here, we show how this limitation may be rectified. First, we identify several theoretical conditions that control paths for Neural CDEs should satisfy, such as boundedness and uniqueness. Second, we use these to motivate the introduction of new schemes that address these conditions, offering in particular measurability (for online prediction), and smoothness (for speed). Third, we empirically benchmark our online Neural CDE model on three continuous monitoring tasks from the MIMIC-IV medical database: we demonstrate improved performance on all tasks against ODE benchmarks, and on two of the three tasks against SOTA non-ODE benchmarks.

## 1 Introduction

Neural differential equations are an elegant formulation combining continuous-time differential equations with the high-capacity function approximation of neural networks. This makes them an appealing methodology for handling irregular time series. Recent examples include Rubanova et al. (2019); Jia & Benson (2019); De Brouwer et al. (2019); Kidger et al. (2020); Herrera et al. (2021); Morrill et al. (2021); Kidger et al. (2021b) amongst others. A comprehensive overview of the subject can be found in Kidger (2022).

Our particular focus is the Neural Controlled Differential Equation (Neural CDE) of Kidger et al. (2020). These were introduced as the general continuous-time limit of arbitrary RNNs. Besides these appealing theoretical connections – and indeed recent work on RNNs has often explicitly designed them around differential-equation-like structures (Chang et al., 2019) – Neural CDEs have additionally been shown to demonstrate excellent empirical performance. In particular, they have been shown to outperform similar Neural ODE or RNN models at modelling functions of irregular time series in offline prediction tasks, where all data is observed in advance (Kidger et al., 2020; Bellot & van der Schaar, 2021).

However despite these appealing properties, Neural CDEs cannot yet be used to learn and predict in real-time (where new data arrives during inference), due to the solution trajectory exhibiting dependence on future data. In contrast, other Neural ODE variants such as the ODE-RNN (Rubanova et al., 2019) can already handle online processing.

In this work, we show how this may be rectified, so that Neural CDEs may be adapted to apply to online problems.

## 1.1 Neural controlled differential equations

Suppose that we observe some time series $\mathbf{x} = \big((t_0, x_0), \ldots, (t_n, x_n)\big)$ with $t_i \in \mathbb{R}$ denoting the timestamp of the observation vector $x_i \in (\mathbb{R} \cup \{*\})^v$, where $*$ is used to denote that some information may be missing, and $t_0 < \ldots < t_n$. Let $X_{\mathbf{x}} \colon [0, n] \to \mathbb{R}^v$ be a (continuous, bounded variation) interpolation such that $X_{\mathbf{x}}(i) = (t_i, x_i)$, where '=' denotes equality up to missing data. We refer to $X_{\mathbf{x}}$ as the control path.[1]

If timestamps are irregular or data are missing, then the frequency of observations may carry information, which simple interpolation would obscure. This is well known to be true of medical ICU data (Che et al., 2018). In such cases, we can replace each $(t_i, x_i) \mapsto (t_i, x_i, c_i)$ where $c_i \in \mathbb{N}_0^v$ counts the number of times the channels in $x_i$ have been observed up to $t_i$. Let $f_{\theta_1} \colon \mathbb{R}^w \to \mathbb{R}^{w \times v}$ and $\zeta_{\theta_2} \colon \mathbb{R}^v \to \mathbb{R}^w$ be neural networks depending on learnable parameters $\theta_1, \theta_2$. Here $w$ is a hyperparameter that describes the size of the hidden state and corresponds to the dimension of the information propagated along the solution trajectory.

Provided $X_{\mathbf{x}}$ is piecewise continuously differentiable (as will always be the case for us), then the Neural CDE model is defined as the solution $z$ to

$$z(t_0) = \zeta_{\theta_2}(t_0, x_0), \quad z(t) = z(t_0) + \int_{t_0}^{t} f_{\theta_1}(z(s)) \frac{\mathrm{d}X_{\mathbf{x}}}{\mathrm{d}s} \mathrm{d}s \quad \text{for } t \in (t_0, t_n], \tag{1}$$

and as such, the model can be interpreted and solved as an ordinary differential equation. Here "$f_{\theta_1}(z(s)) \frac{\mathrm{d}X_{\mathbf{x}}}{\mathrm{d}s}$" denotes a matrix-vector product. The solution $z$ is said to be the response of a CDE *driven or controlled by* $X_{\mathbf{x}}$.

The evolving $z(t) \in \mathbb{R}^v$ is analogous to the hidden state in an RNN, now operating in continuous time. Typically, the output of the model will be a linear map on this hidden state: either applied to $z(t)$ for all times $t \in [t_0, t_n]$ to produce a time-dependent output path, or on just $z(t_n)$ for a single output such as for classification.

## 1.2 Continuous time control signals for Neural CDEs

Neural CDEs act on and require a continuous-time embedding $X_{\mathbf{x}}$ of the observed data $\mathbf{x}$. This provides a number of benefits. Firstly, this simplifies the handling of 'messy' (irregularly sampled with missing data) time series, by enabling it to be interpreted in the same way as regular data. Additionally, as this results in an ODE-like model, the model produces a continuously defined solution, has memory-efficient continuous adjoint methods, and the utilisation of modern ODE solvers offers trade-offs between error and computation.

Current implementations take the map $\mathbf{x} \to X_{\mathbf{x}}$ to be either a natural cubic spline or a linear interpolation (Kidger et al., 2020; Morrill et al., 2021), of which neither can be used in an online fashion. This is because any segment of the control path depends on all data values, and therefore predictions at time $t_i$ depend on future, as-yet-unobserved, data at $t > t_i$.

## 1.3 Contributions

First, we formalise the requirements for what it means to be an ideal Neural CDE control path by introducing four theoretical conditions that it should satisfy: *adapted measurability*, *smoothness*, *boundedness*, and *uniqueness*.

---

[1] We can actually have $X_{\mathbf{x}} \colon [s_0, s_n] \to \mathbb{R}^v$ with $X_{\mathbf{x}}(s_i) = (t_i, x_i)$ for any $s_0 < \cdots < s_n$, as the reparameterisation invariance property of CDEs means there is no change in the solution, see Appendix A.2. For example Kidger et al. (2020) took $s_i = t_i$. We prefer $s_i = i$ as it is an easier choice when batching data.

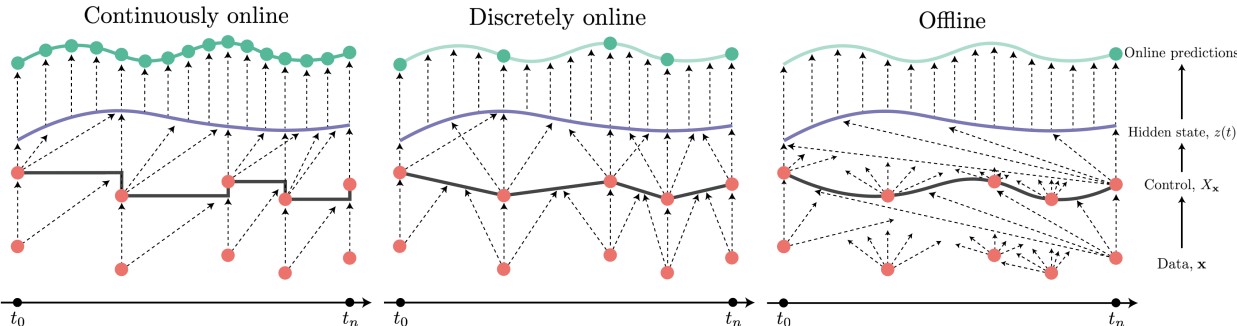

Figure 1: Graphical description of the measurability definition for Neural CDEs. Arrows indicate the direction of time datapoints can influence and ● indicates an online prediction can be made at that point. **Left:** a continuously online model where no information is passed backwards in time, resulting in an online solution at all points in time. **Middle:** a discretely online scheme where information can be passed backwards, but not further than the preceding observation. **Right:** an offline scheme where information is passed backwards in time further than the preceeding observation.

Second, we use these conditions as a guide to introduce two new control signals, namely the *rectilinear control* and *cubic Hermite splines with backward differences*. These are designed to satisfy the previous theoretical conditions, as well as to address the drawbacks of previously considered schemes. In particular, by utilising the invariance of control paths to reparameterisation, the rectilinear control we construct can be used in all online prediction tasks. This enables neural CDEs to be as flexible in application as RNNs.

Third, we show that this expansion in the domain of application does not come at the cost of performance. We verify the empirical behaviour of our new schemes for constructing control paths, and provide straightforward recommendations into which scheme to use when. We run benchmark experiments on three continuous monitoring tasks, drawn from the medical time series MIMIC-IV database. In this regime, we demonstrate that Neural CDEs exhibit improved performance on all tasks against ODE benchmarks, and on two of the three tasks against non-ODE models.

## 2 What makes a good control signal?

We now introduce four conditions that a good control signal should satisfy.

### 2.1 Adapted measurability

A model which can learn and predict in real-time is often referred to as an *online model* in machine learning. This is analogous to the concept of an *adapted measurable process* in the language of probability theory (Williams, 1991). This is the primary property of interest to us here, which is not satisfied by existing implementations of Neural CDEs. Fulfilment of this property is what will enable Neural CDEs to be deployed in real-world real-time scenarios, such as continuous monitoring in ICU settings.

$\mathcal{T}-$**measurable**   Let $\mathcal{T} \subseteq [t_0, t_n]$. We say that the Neural CDE solution $z(t)$ (of Equation (1)) is $\mathcal{T}-$measurable if for all $t \in \mathcal{T}$ we have that $z(t)$ is a function of only observations $(t_i, x_i)$ for those $t_i \in [t_0, t]$. That is, for all $t \in \mathcal{T}$ it is possible to obtain a prediction in an online setting.

**Continuously online**   If the solution is $\mathcal{T}-$measurable for $\mathcal{T} = [t_0, t_n]$, then we say that it is *continuously online*. That is, an online prediction task can be defined for any time, and the existing control path will not be affected by new observations. The incorporation of new data at $t_{n+1}$ means that we update $z(t)$ for $t \geq t_{n+1}$ (with $z(t)$, $t < t_{n+1}$ unchanged).

**Discretely online**   Suppose that the solution $z(t)$ is $\mathcal{T}-$measurable for $\mathcal{T} = \{t_0, \ldots, t_n\}$ (that is, only at the observation times). For times between the observation times, $t$ in $(t_i, t_{i+1})$, suppose $z(t)$ may also

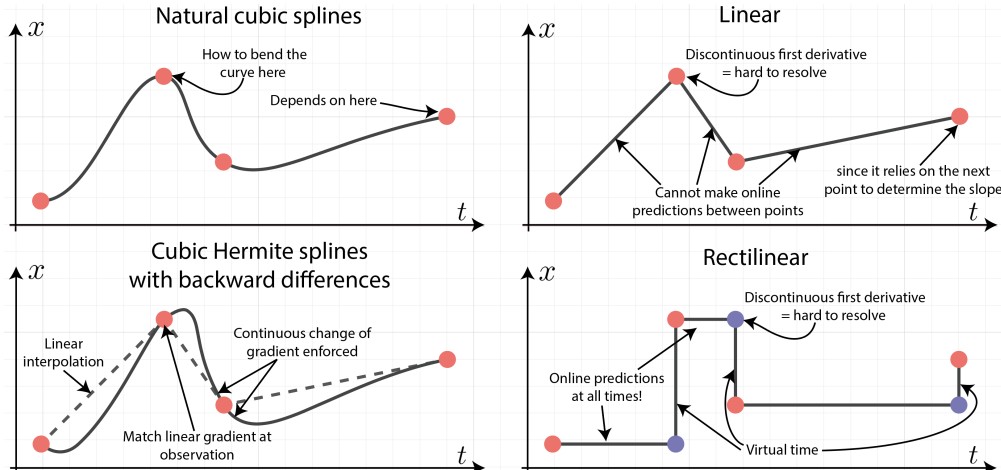

Figure 2: Graphical comparison of the four control signals: natural cubic splines (top left), linear control (top right), cubic Hermite splines with backward differences (bottom left), rectilinear control (bottom right).

depend on the one-step ahead observation, so $z(t)$ is a function of $((t_0, x_0), \ldots, (t_{i+1}, x_{i+1}))$. Then we call the solution *discretely online*. In other words, $z(t)$ is online only at the discrete observation times and the incorporation of the next observation involves recomputing $z(t)$ for $t > t_n$.

We note that if there are missing data then this will usually destroy the discretely online property. To see why this is the case, consider the following data observations

$$\mathbf{x} = \big((t_0, x_0), (t_1, *), (t_2, x_2)\big), \tag{2}$$

with $*$ denoting missing data. The control at time $t_1$ is dependent on the value $x_2$ at $t_2$, and as such, it cannot define a discretely online solution at $t_1$.

**Offline**  If the solution is not at least discretely online then we say it is *offline*. This means that in prediction tasks, the arrival of a new data point will cause changes in the control path such that $z(t)$ has to be recomputed for all $t$.

We choose to sub-categorise the definitions in this way since it well separates existing models. For example, standard RNNs are discretely online, an ODE-RNN (Rubanova et al., 2019) is continuously online, and existing Neural CDEs are offline.

A graphical depiction of this is shown in Figure 1.

## 2.2  Smoothness

To be able to apply numerical solvers to the integral in Equation (1) we require that the integrand be sufficiently smooth, and thus that the control be sufficiently smooth. As a minimum (using Euler's method), we require that $X_{\mathbf{x}}$ be piecewise twice continuously differentiable with bounded second derivative. In practice to use a higher-order numerical method, such as Dormand–Prince, then additional smoothness is desirable.

When using an adaptive solver and a control that is piecewise smooth, but not smooth, the solver should be informed about the jumps between pieces so that its integration steps aligns with them. Without this the solver must locate the discontinuities on its own, slow down to resolve them, and then speed up again, which numerically expensive. For example this may be done using the `jump_t` argument for `torchdiffeq` (Chen et al., 2018), or the `d_discontinuities` argument for DifferentialEquations.jl (Rackauckas & Nie, 2017).

### 2.3 Boundedness

We require that $X_{\mathbf{x}}$ should behave "reasonably". It should not introduce any spurious oscillations or grow unboundedly from bounded data. This condition ensures that $X_{\mathbf{x}}$ is a good representative of $\mathbf{x}$, so that the Neural CDE can learn from the data.

Formally, let $\tau_i = t_{i+1} - t_i$ for each $i$. Then we require that there exists some continuous $\omega \colon \mathbb{R} \times \mathbb{R} \times \mathbb{R} \to \mathbb{R}$ such that

$$\|X_{\mathbf{x}}\|_{\infty} + \left\|\frac{\mathrm{d}X_{\mathbf{x}}}{\mathrm{d}t}\right\|_{\infty} + \left|\frac{\mathrm{d}X_{\mathbf{x}}}{\mathrm{d}t}\right|_{BV} < \omega(\max_i \tau_i, \min_i \tau_i, \max_i |x_i|), \tag{3}$$

where $|\cdot|_{BV}$ denotes the bounded variation seminorm.

This condition is required to attain the universal approximation property for Neural CDEs. A formal proof is given in Appendix B, but the main idea is as follows. Consider a collection of time series $\mathcal{X}$ for which $\sup_{\mathbf{x} \in \mathcal{X}} \omega(\max_i \tau_i, \min_i \tau_i, \max_i |x_i|) < \infty$. Then (3) implies that $\mathfrak{X} = \{X_{\mathbf{x}} \,|\, \mathbf{x} \in \mathcal{X}\}$ is relatively compact with respect to the topology generated by $\|X\|_{\infty} + \|\mathrm{d}X/\mathrm{d}t\|_1$, and hence also with respect to the topology generated by $\|X\|_{\infty} + |X|_{BV}$. As this then satisfies the compactness condition of Kidger et al. (2020, Theorem B.7), Neural CDEs driven by $X \in \mathfrak{X}$ are universal approximators on $\mathbf{x} \in \mathcal{X}$.

We note that this property is non-trivial: for example, quadratic splines exhibit a resonance property that may result in unbounded oscillations as time progresses.

### 2.4 Control signal uniqueness

Given a collection of time series $\mathcal{X}$, we say that a control $X_{\mathbf{x}}$ is unique with respect to $\mathcal{X}$ if

$$\mathbf{x} \to (x_0, \mathrm{Signature}(X_{\mathbf{x}})) \text{ is injective with respect to } \mathcal{X}. \tag{4}$$

This is required, along with the boundedness property, for universal approximation of Neural CDEs to hold. Signature denotes the signature transform, which is central to the study of CDEs (Lyons, 1998; Lyons et al., 2007; Bonnier et al., 2019; Morrill et al., 2020; Kidger & Lyons, 2021; Morrill, 2022).

For the reader unfamiliar with signature transforms, this may intuitively be approximated by the simplified (but not completely accurate) condition

$$\mathbf{x} \to X_{\mathbf{x}} \text{ is injective with respect to } \mathcal{X}. \tag{5}$$

That is, we require every possible unique set of data to have a unique control path.

If the dataset is regularly sampled, then (5) is immediately satisfied (by the $X_{\mathbf{x}}(i) = (t_i, x_i)$ property). If the dataset is irregularly sampled, then this may fail. One way to recover (5) is to include the observational frequencies $c_i$. See Appendix B.2.

## 3 Control signals for Neural CDEs

As mentioned in Section 2, previous work has not explored the choice of control in much detail. Here we overview existing controls and identify the theoretical properties from Section 2 that each possesses. We then introduce two new controls – cubic Hermite splines with backward differences, and rectilinear controls – that address issues with existing control schemes.

**Natural cubic splines** Natural cubic splines were used in the original Neural CDE paper by Kidger et al. (2020). This control signal requires the full time series to be available prior to construction, as $z(t)$ depends on all datapoints $\big((t_0, x_0), \ldots (t_n, x_n)\big)$. Any change in one datapoint has a small effect on the entire construction, even at earlier times. As such, natural cubic splines cannot be used in an online fashion. If an offline scheme is sufficient, then these do still make a good choice: they are relatively smooth and slowly varying, making them fast to integrate numerically.

| Control signal | Properties | | | |
| | Measurability | Smoothness | Boundedness | Uniqueness |
|---|---|---|---|---|
| Natural cubic | ✗ | ✔ | ✔ | $\sim$ |
| Linear | (discrete)* | (piecewise) | ✔ | ✔(with $c_i$) |
| Cubic Hermite | (discrete)* | ✔ | ✔ | ✔(with $c_i$) |
| Rectilinear | ✔ | (piecewise) | ✔ | ✔(with $c_i$) |

Table 1: Summary of the interpolation schemes and the properties they hold for irregular and partially observed data. A superscript $*$ denotes that the property holds only if no data is missing, and $\sim$ if the property is unknown. Proofs for previously unknown properties are given in Appendix B
.

**Linear control**  This is arguably the simplest and most intuitive control signal, whereby we apply linear interpolation between observations. For fully observed data, the linear control defines a discretely online control path, and thus has the same online properties as an RNN.

Whilst it has better online properties, the linear control is generally slower than natural cubic splines. This is due to the need to resolve derivative discontinuities at the knots $(t_i, x_i)$, as mentioned in Section 2.2.

**Cubic Hermite splines with backward differences**  This scheme smooths the discontinuities in the linear control, whilst retaining the same online properties. We achieve this by joining adjacent timepoints with a cubic spline where the additional degrees of freedom are used to smooth gradient discontinuities. This leads to faster integration times than linear controls.

This differs from natural cubic splines as it solves a single equation on each $[i, i+1)$ piece independently. As a result, it is more quickly varying than natural cubic splines (see Figure 2) and so produces slower integration times than natural cubic splines.

For each interval $[i, i+1)$, we ensure that $X_{\mathbf{x}}(i) = (t_i, x_i)$, $X_{\mathbf{x}}(i+1) = (t_{i+1}, x_{i+1})$, and that the gradient at each node matches the backward finite difference

$$\frac{\mathrm{d}X_{\mathbf{x}}}{\mathrm{d}t}(i) = x_i - x_{i-1},$$
$$\frac{\mathrm{d}X_{\mathbf{x}}}{\mathrm{d}t}(i+1) = x_{i+1} - x_i.$$

The result of this is a control signal with continuous derivatives that is discretely online.

**Rectilinear control**  Each of the previous schemes need to look at least one time-step ahead to construct the control between time points, so they are not continuously online.

To resolve this issue, we propose the following procedure. Let $\widetilde{x}_i$ denote the forward fill of $x_i$ with respect to the missing data $*$. We then let $X_{\mathbf{x}} \colon [0, 2n] \to \mathbb{R}^v$ be piecewise linear such that $X_{\mathbf{x}}(2i) = (t_i, \widetilde{x}_i, c_i)$ for $i \in \{0, \ldots, n\}$ and $X_{\mathbf{x}}(2i-1) = (t_{i+1}, \widetilde{x}_i, c_i)$ for $i \in \{1, \ldots, n\}$, where $c_i$ is the observational frequency.

This produces a control signal – the *rectilinear control* – that updates the time and feature channels separately in a lead-lag fashion. While channels are not observed, time is increased as normal. When a channel is observed, we interpolate between channel values with the real time held fixed. We term the interpolation between channel values the *virtual time*, since real time is fixed but the parameterisation is moved forward. We have converted the time parameterisation onto a space-time parameterisation. This is shown pictorially in the bottom right of Figure 2.

Note that this step is analogous to the concept of innovation in filtering/signal processing problems (Bain & Crisan, 2009). ODE-RNNs (Rubanova et al., 2019) may in fact be seen as a special case of a Neural CDE with a rectilinear control.

Evolving the neural CDE along the virtual time at the instance data arrives enables us to update our solution at the instant data arrives before continuing with the standard time evolution. This methodology has hitherto

Tables 2 to 5: Control signal comparison for the *regularly sampled* datasets using the 'dopri5' adaptive solver. The top performer for each dataset is given in bold. NFEs represents the number of function evaluations per epoch.

| Control | RMSE | NFEs $\times 10^3$ |
|---|---|---|
| Natural cubic | $53.3 \pm 0.3$ | $\mathbf{4.6 \pm 0.1}$ |
| Linear | $\mathbf{51.7 \pm 0.8}$ | $5.4 \pm 0.3$ |
| Cubic Hermite | $52.5 \pm 0.7$ | $4.8 \pm 0.1$ |
| Rectilinear | $52.0 \pm 1.3$ | $16.3 \pm 0.4$ |

Table 2: **BeijingPM2.5**

| Control | RMSE | NFEs $\times 10^3$ |
|---|---|---|
| Natural cubic | $77.6 \pm 1.7$ | $\mathbf{4.5 \pm 0.1}$ |
| Linear | $\mathbf{77.5 \pm 2.4}$ | $5.7 \pm 0.1$ |
| Cubic Hermite | $78.5 \pm 1.6$ | $4.7 \pm 0.1$ |
| Rectilinear | $79.0 \pm 0.7$ | $15.7 \pm 0.4$ |

Table 3: **BeijingPM10**

| Control | ACC (%) | NFEs $\times 10^3$ |
|---|---|---|
| Natural cubic | $83.6 \pm 6.1$ | $\mathbf{1.0 \pm 0.2}$ |
| Linear | $97.6 \pm 1.5$ | $2.2 \pm 0.1$ |
| Cubic Hermite | $\mathbf{99.3 \pm 0.0}$ | $1.9 \pm 0.0$ |
| Rectilinear | $98.6 \pm 0.5$ | $7.9 \pm 0.4$ |

Table 4: **CharacterTrajectories**

| Control | ACC (%) | NFEs $\times 10^4$ |
|---|---|---|
| Natural cubic | $92.9 \pm 0.3$ | $\mathbf{2.67 \pm 0.36}$ |
| Linear | $93.3 \pm 0.7$ | $6.35 \pm 0.50$ |
| Cubic Hermite | $92.5 \pm 0.4$ | $4.16 \pm 0.03$ |
| Rectilinear | $\mathbf{93.7 \pm 0.8}$ | $11.3 \pm 0.67$ |

Table 5: **SpeechCommands**

not been used for neural CDEs and is what enables the model to be used in online prediction tasks. The downside of this scheme is that the parameterisation is twice as long (domain is $[0, 2n]$ rather than $[0, n]$), with correspondingly many derivative discontinuities. This means that the model takes longer to evaluate, and to train.

In Table 1, we summarise each of the aforementioned schemes, and the properties from Section 2 that each holds. Corresponding proofs are given in Appendix B.

# 4 Experiments

We start by illustrating the effect of using an offline method on a simulated toy problem to predict whether a Brownian path terminates above zero. We then evaluate each of the controls across a range of both regularly sampled and irregularly sampled datasets. Finally, we benchmark the online Neural CDE against a collection of benchmarks on MIMIC-IV prediction tasks.

Apart from the toy example, each model used hyperparameters that were chosen by a Bayesian optimisation strategy with 20 trials. The ranges of values the hyperparameters were optimised over as well as the final configuration for every model is given in Appendix C. Full experimental details including optimisers, learning rates, normalisation, architectures and so on can be found in Appendix C. The code for reproducing all results is available at [redacted for anonymity].

## 4.1 Simulated data prediction problem

We construct a toy example to demonstrate why the online property warrants particular emphasis. Simulating standard Brownian paths starting at the origin, we predict whether the paths will be $> 0$ at $t = 1$ at prediction times $\{0, 0.5, 1\}$.

Theoretically, given information up to the current time (this is Markov by independent Gaussian increments), an online prediction cannot do better than 75% accuracy. However, as we see in Table 6, if the natural cubic spline is used to construct the control path, then there is some inevitable information bleed, therefore we obtain an accuracy of 81.7%. It is precisely this feature/bug that makes the previous implementation of Neural CDEs unsuitable for online problems. Further details can be found in Appendix C.1.

| Control | ACC(%) |
|---|---|
| Natural cubic | **81.7 ± 0.007** |
| Linear | 75.0 ± 0.003 |
| Cubic Hermite | 74.8 ± 0.006 |
| Rectilinear | 74.3 ± 0.006 |

Table 6: Predicting location of Brownian path.

## 4.2 Practical Datasets

We now overview the datasets used in our analysis. These all had a 70%/15%/15% train/validation/test split, with (as required by the Neural CDE formulation) time included as a channel.

We begin with four regularly sampled and fully observed datasets.

**BeijingPM2.5, BeijingPM10:** Both contain 10 channels and 16966 total samples. The aim is to predict the PM2.5/PM10 level (two pollutant indexes) at 12 different air-quality monitoring sites in Beijing (Tan et al., 2020). The performance metric is taken to be the RMSE against the true value.

**SpeechCommands:** Contains 11 channels and 34975 samples of one-second audio recordings of individual spoken words such as 'yes', 'no', 'left', and 'right' (Warden, 2018). The performance metric is taken to be classification accuracy of the spoken words.

**CharacterTrajectories:** Contains 4 channels and 2872 samples of the $x, y$ position and the force applied by the pen tip whilst writing a letter from the Latin alphabet in a single stroke (Bagnall et al., 2018). The performance metric used is the classification accuracy for the different characters.

In addition to these, we additionally examine three tasks for the MIMIC-IV database (Johnson et al., 2021; Goldberger et al., 2000) related to continuous patient health monitoring. The database contains 76540 de-identified admissions to intensive care units at the Beth Israel Deaconess Medical Center. The data is highly irregular and channels have lots of missing data. Each of the three tasks has its own set of exclusion criteria; these are outlined in full in Appendix C.4.

**Mortality:** We predict the likelihood of mortality within an ICU stay from some initial data. The performance metric is the AUC of prediction of eventual mortality.

**LOS:** We estimate the length of stay of a patient given their first 24 hours of data. The performance metric is the RMSE against the true value in days.

**Sepsis:** We predict the risk of sepsis along a patient's stay. The performance metric is the AUC against the true state of sepsis that is defined according to the Sepsis-3 definition (Singer et al., 2016) (full details are given in Appendix C.4).

In terms of metrics, lower is better for BeijingPM2.5, BeijingPM10, and LOS, whilst higher is better for the others.

## 4.3 Empirical study on control signals

In Tables 2 to 5 we compare the performance of the control signals on the regularly sampled datasets.

The NFEs column (Number of Function Evaluations per epoch) shows that natural cubic splines are uniformly the fastest choice. For example, on the SpeechCommands dataset, natural cubic splines require an average of $2.55 \times 10^4$ evaluations per epoch, piecewise cubic nearly double on $4.16 \times 10^4$, linear nearly triple on $6.35 \times 10^4$, and rectilinear significantly more on $1.08 \times 10^5$.

However, the linear, cubic Hermite, and rectilinear schemes all produce better RMSE and accuracies, generally similar to each other. As they are the fastest from this group, we thus recommend cubic Hermite splines with backward differences as the best choice on regular datasets.

Tables 7 and 8: Control signal comparison for the *irregularly sampled* Mortality and LOS MIMIC-IV tasks using the 'dopri5' adaptive solver. The top performer for each dataset is given in bold. NFEs represents the number of function evaluations per epoch.

| Control | AUC | NFEs$\times 10^3$ |
|---|---|---|
| Natural cubic | $0.859 \pm 0.003$ | $\mathbf{14.5 \pm 0.1}$ |
| Linear | $\mathbf{0.910 \pm 0.003}$ | $36.9 \pm 0.8$ |
| Cubic Hermite | $0.909 \pm 0.002$ | $26.0 \pm 0.6$ |
| Rectilinear | $0.906 \pm 0.002$ | $96.3 \pm 0.1$ |

Table 7: **Mortality**

| Control | RMSE | NFEs$\times 10^3$ |
|---|---|---|
| Natural cubic | $0.241 \pm 0.005$ | $\mathbf{4.0 \pm 0.0}$ |
| Linear | $0.149 \pm 0.037$ | $7.8 \pm 0.2$ |
| Cubic Hermite | $0.138 \pm 0.025$ | $4.2 \pm 0.1$ |
| Rectilinear | $\mathbf{0.109 \pm 0.008}$ | $11.9 \pm 1.0$ |

Table 8: **LOS**

| | MIMIC-IV | | |
|---|---|---|---|
| **Model** | Mortality | LOS | Sepsis |
| GRU | $0.846 \pm 0.05$ | $0.236 \pm 0.028$ | $0.791 \pm 0.002$ |
| GRU-dt | $0.912 \pm 0.003$ | $0.149 \pm 0.008$ | $0.8 \pm 0.001$ |
| GRU-dt-intensity | $0.924 \pm 0.014$ | $0.142 \pm 0.013$ | $\mathbf{0.801 \pm 0.002}$ |
| GRU-D | $0.93 \pm 0.002$ | $0.148 \pm 0.002$ | $\mathbf{0.801 \pm 0.002}$ |
| ODE-RNN | $0.524 \pm 0.004$ | $0.154 \pm 0.004$ | $0.793 \pm 0.001$ |
| Online Neural CDE | $0.908 \pm 0.003$ | $0.11 \pm 0.009$ | $0.77 \pm 0.002$ |
| Online Neural CDE w/ observational freq | $\mathbf{0.943 \pm 0.003}$ | $\mathbf{0.099 \pm 0.001}$ | $0.795 \pm 0.004$ |

Table 9: Benchmarking online Neural CDEs (Neural CDEs with rectilinear interpolation) using the 'rk4' solver with and without observational frequency against a range of algorithms on the MIMIC-IV tasks. The top performer for each problem is shown in bold.

We see a similar story on the irregularly sampled MIMIC-IV tasks in Tables 7 and 8 (see also Appendix C.4 for results on the sepsis task). Natural cubic splines are again the fastest, but again their AUC/RMSE performance is poor. Linear/piecewise cubic performs best on the Mortality prediction tasks, whereas rectilinear is the best on LOS. Note that only the rectilinear model can actually be deployed in a continuous ICU monitoring scenario.

## 4.4 Benchmarking on MIMIC-IV

Finally, we benchmark the online (rectilinear) Neural CDE on the three tasks from the MIMIC-IV database.

We highlight that Neural CDEs have been tested before on such problems before (notably Kidger et al. (2020) on a sepsis detection tasks); however, this is the first such benchmarking of a Neural CDE model that can actually be deployed in an *online, real-time*, in-hospital environment.

Our benchmarks include: vanilla GRU; GRU-dt, which is a GRU that additionally includes the time difference between observations; GRU-dt-intensity, the same as a GRU-dt but also includes the observational intensity; GRU-D (Che et al., 2018), incorporates both time differences and observational intensity but in a more sophisticated manner; and ODE-RNN (Rubanova et al., 2019). The ODE-RNN is chosen as it represents a continuously online ODE-benchmark, whereas all other models operate only discretely online.

The results over three runs are presented in Table 9. We see that upon inclusion of the measurement intensity matrix, the causal Neural CDE becomes extremely competitive with the GRU-D benchmark, achieves convincing improvements in performance in the LOS and mortality tasks, and is only narrowly beaten in the sepsis detection task. We also see across-the-board superior performance to the ODE-RNN.

It is already known that Neural CDEs are effective at modelling irregular functions on time series. However, these results, in particular those on rectilinear controls, are the first demonstration that Neural CDEs can be adapted for use in a real-world online scenario whilst retaining performance.

## 5 Related work

Several works have now studied Neural CDEs in some capacity.

Bellot & van der Schaar (2021) develop a SOTA counterfactual estimation method in continuous time for irregular data using the Neural CDE. They state that misaligned observation times had been a problem for previous methods, and none had been able to operate in continuous time.

Morrill et al. (2021) apply techniques from rough path theory so that Neural CDEs may better handle very long time series.

Zhuang et al. (2021) apply a reversible ODE solver so that both optimise-then-discretise and discretise-then-optimise methods produce the same gradients. Meanwhile Kidger et al. (2021a) tweak the numerical solver to approximately double the speed of training Neural CDEs via optimise-then-discretise methods.

More broadly, there has been much interest in applying neural differential equations to time series.

Rubanova et al. (2019) introduced ODE-RNNs, which are a neural jump ordinary differential equation with jumps at each observation. Meanwhile De Brouwer et al. (2019); Herrera et al. (2021) take very similar approaches to each other, using neural ODEs to perform continuous-time filtering.

Jia & Benson (2019); Li et al. (2020); Kidger et al. (2021b) amongst others consider Neural SDEs for time series modelling. The distinction is that Neural CDEs are used to model functions of time series (typically supervised learning), whilst Neural SDEs seek to model the time series themselves (typically unsupervised learning).

Other neural differential equation based unsupervised approaches to time series modelling use random ODEs (Norcliffe et al., 2021) and spatio-temporal point processes (Chen et al., 2021b).

Several works have studied using neural differential equations to model time series arising from physical systems, for example chemical kinetic modelling (Kim et al., 2021), oscillatory dynamical systems (Norcliffe et al., 2020), systems with switching behaviour (Chen et al., 2021a), and those arising in general scientific modelling (Rackauckas et al., 2020).

## 6 Discussion and limitations

**Novelty**

1. The extension of Neural CDEs to online problems is a **qualitative step-change in their applicability**. Whereas they have previously been restricted only to offline problems, they may now be applied as flexibly as RNNs, which are their discrete-time counterparts.

2. The experimental results demonstrate significant accuracy/loss improvements on all datasets over the previous (natural cubic spline) implementation, **even for offline tasks**.

**Recommendations for the control signal**  Considering the results from Section 4 and the properties from Section 2, we make the following recommendations for Neural CDE control paths:

1. If the problem is online and requires the solution to be continuously online, then use rectilinear controls. This is the only scheme that is continuously online.

   Likewise if the problem is online, has missing data, and requires the solution to be discretely online, then use rectilinear controls. This is the only scheme that is discretely online in the presence of missing data.

2. If the problem is online, has no missing data, and requires the solution to be discretely online, then use cubic Hermite splines with backward differences. We see in Tables 2 to 5, 7 and 8 that the performance is in-line with both linear and rectilinear, but at faster speeds.

   Likewise, these are also recommended if the problem is offline.

3. If speed is of greater importance than accuracy, and the task is offline, then use natural cubic splines. This is evidenced by the results in Tables 2 to 5 which show natural cubic splines to be significantly faster than other alternatives.

As such our findings recommend either rectilinear control or cubic Hermite splines with backward differences, both introduced in this paper, for most cases.

**Limitations**  The main limitation of the techniques introduced here come from rectilinear controls. These are typically slow, and if trained with discretise-then-optimise techniques come with high memory usage. Despite this, in many online cases these are 'the only game in town'.

**Implementation**  To help facilitate adoption, both rectilinear controls and cubic Hermite splines with backward differences have been implemented in the `torchcde` open-source library for CDEs.

## 7  Conclusion

We formalised the properties that ideal Neural CDE control schemes should have. In doing so, we identified two new control schemes that address issues with existing implementations, in particular with respect to online predictions and speed. Having performed both a theoretical and empirical study into the schemes' behaviour, we provide recommendations regarding which scheme to use when. This included benchmarking the online Neural CDE on three continuous monitoring ICU tasks, in which improved and state-of-the-art performance is demonstrated against similar ODE or RNN based approaches.

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

# A    Neural controlled differential equations

## A.1    RNNs are a special case of Neural CDEs

This is easily shown by noting that an RNN of the form

$$z_{i+1} = h_\theta(z_i, x_i)$$

is an explicit Euler discretisation with step unit length of

$$z(t) = z(t_0) + \int_{t_0}^t h_\theta\big(z(s), X_{\mathbf{x}}(s)\big) - z(s) \, \mathrm{d}s,$$

which is a special case of a Neural CDE (Kidger et al., 2020, Theorem C.1).

## A.2    Reparameterisation invariance of Neural CDEs

CDEs exhibit a *reparameterisation invariance* property.[2]

Let $\psi \colon [a,b] \to [c,d]$ be differentiable, increasing, and surjective, with $\psi(a) = c$ and $\psi(b) = d$. Let $T \in [c,d]$, let $\hat{z} = z \circ \psi$, let $\hat{X} = X \circ \psi$, and let $S = \psi(T)$. Then substituting $t = \psi(\tau)$ into a CDE (and using a standard change of variables):

$$\hat{z}(S) = z(T) \tag{6}$$

$$= z(c) + \int_c^T f(z(t)) \mathrm{d}X(t) \tag{7}$$

$$= z(c) + \int_c^T f(z(t)) \frac{\mathrm{d}X}{\mathrm{d}t}(t) \mathrm{d}t \tag{8}$$

$$= z(\psi(a)) + \int_a^{\psi^{-1}(T)} f(z(\psi(\tau))) \frac{\mathrm{d}X}{\mathrm{d}t}(\psi(\tau)) \frac{\mathrm{d}\psi}{\mathrm{d}\tau}(\tau) \mathrm{d}\tau \tag{9}$$

$$= (z \circ \psi)(a) + \int_a^{\psi^{-1}(T)} f((z \circ \psi)(\tau))) \frac{\mathrm{d}(X \circ \psi)}{\mathrm{d}\tau}(\tau) \mathrm{d}\tau \tag{10}$$

$$= (z \circ \psi)(a) + \int_a^{\psi^{-1}(T)} f((z \circ \psi)(\tau)) d(X \circ \psi)(\tau) \tag{11}$$

$$= \hat{z}(c) + \int_c^S f(\hat{z}(\tau)) d\hat{X}(\tau) \tag{12}$$

From this we can see that $\hat{z}$ satisfies the Neural CDE equation, now with $\hat{X}$ as the control.

# B    Theoretical conditions on the control

## B.1    Boundedness

### B.1.1    Boundedness is required for universal approximation

Let $\mathcal{X}$ be some set of time series such that

$$\sup_{\mathbf{x} \in \mathcal{X}} \omega(\max_i \tau_i, \min_i \tau_i, \max_i |x_i|) < \infty.$$

Let $\mathfrak{X} = \{X_{\mathbf{x}} \mid \mathbf{x} \in \mathcal{X}\}$. By equation (3), then

$$\sup_{X \in \mathfrak{X}} \|X\|_\infty + \left\| \frac{\mathrm{d}X}{\mathrm{d}t} \right\|_\infty + \left| \frac{\mathrm{d}X}{\mathrm{d}t} \right|_{BV} < \infty.$$

---

[2]In fact, they also exhibit a tree-like invariance property Hambly & Lyons (2010), which is a slight generalisation.

Let $\mathfrak{X}' = \{dX/dt \mid X \in \mathfrak{X}\}$. Then $\mathfrak{X}$ is bounded in $W^{1,\infty}[t_0, t_n]$ and so relatively compact in $L^\infty[t_0, t_n]$, and $\mathfrak{X}'$ is bounded in $BV[t_0, t_n]$ and so relatively compact in $L^1[t_0, t_n]$. Therefore $\mathfrak{X} \times \mathfrak{X}'$ is relatively compact in $L^\infty[t_0, t_n] \times L^1[t_0, t_n]$.

Let $\mathbb{X} = \{(X, dX/dt) \mid X \in \mathfrak{X}\}$. Then $\mathbb{X} \subseteq \mathfrak{X} \times \mathfrak{X}'$ so $\mathbb{X}$ is also relatively compact in $L^\infty[t_0, t_n] \times L^1[t_0, t_n]$. This implies that $\mathfrak{X}$ is relatively compact with respect to the topology generated by $\|X\|_\infty + \|dX/dt\|_1$, and hence also with respect to the topology generated by $\|X\|_\infty + |X|_{BV}$.

This now satisfies the compactness condition of Kidger et al. (2020, Theorem B.7), implying that Neural CDEs driven by $X \in \mathfrak{X}$ are universal approximators on $\mathbf{x} \in \mathcal{X}$.

### B.1.2 Boundedness of cubic Hermite splines with backward differences

The boundedness result for natural cubic splines is given in Kidger et al. (2020, Lemma B.9). We now give a corresponding proof for cubic Hermite splines with backward differences.

Let $X_\mathbf{x} \colon [t_0, t_n] \to \mathbb{R}$ be the cubic Hermite splines with backward differences such that $X_\mathbf{x}(t_i) = x_i$. Note that it is sufficient to consider $X_\mathbf{x} \in \mathbb{R}$ here since each dimension is interpolated separately. Let the $i^{\text{th}}$ piece of $X_\mathbf{x}$, on the interval $[t_i, t_{i+1}]$ be denoted by $X_i$. Without loss of generality, translate each piece onto the interval $[0, \tau_i]$ where $\tau_i = t_{i+1} - t_i$, so that $X_i \colon [0, \tau_i] \to \mathbb{R}$. Let $X_i(t) = a_i + b_i t + c_i t^2 + d_i t^3$ for some coefficients $a_i, b_i, c_i, d_i$ and $i \in \{0, \ldots, n-1\}$.

For cubic Hermite splines with backward differences we enforce $X_i(0) = x_i$, $X_i(\tau_i) = x_{i+1}$, along with the derivative conditions

$$X_i'(0) = \frac{x_i - x_{i-1}}{\tau_{i-1}}, \tag{13}$$

$$X_{i+1}'(\tau_i) = \frac{x_{i+1} - x_i}{\tau_i}. \tag{14}$$

Letting $\Delta x_i = x_i - x_{i-1}$, we find

$$a_i = x_i, \tag{15}$$

$$b_i = \tau_{i-1}^{-1} \Delta x_i, \tag{16}$$

$$c_i = 2\tau_i^{-2} \tau_{i-1}^{-1} (\tau_{i-1} \Delta x_{i+1} - \tau_i \Delta x_i), \tag{17}$$

$$d_i = \tau_i^{-3} \tau_{i-1}^{-1} (\tau_i \Delta x_i - \tau_{i-1} \Delta x_{i+1}). \tag{18}$$

Letting $x_{\max} = \max_i |x_i|$, $\tau_{\max} = \max_i \tau_i$, $\tau_{\min} = \min_i \tau_i$, it can be shown that

$$\|X_\mathbf{x}\|_\infty \leq C_1 \Big(x_{\max} + \frac{x_{\max} \tau_{\max}}{\tau_{\min}}\Big), \tag{19}$$

$$\left\|\frac{dX_\mathbf{x}}{dt}\right\|_\infty \leq C_2 \frac{x_{\max}}{\tau_{\min}}, \tag{20}$$

$$\left|\frac{dX_\mathbf{x}}{dt}\right|_{BV} \leq C_3 \frac{x_{\max}}{\tau_{\min}^2}, \tag{21}$$

$$\tag{22}$$

for fixed constants $C_1, C_2, C_3 \in \mathbb{R}$. This proves the boundedness property from Equation (3) for cubic Hermite splines with backward differences.

## B.2 Uniqueness

### B.2.1 Interpolation schemes are non-unique in general

Here we give a simple counterexample to prove non-uniqueness of the interpolation schemes if only $(t_i, x_i)$, and not $c_i$, are included as information.

Let $\mathbf{x}_1$ and $\mathbf{x}_2$ be two time series such that

$$\mathbf{x}_1 = \big((t_0, x_0), (t_1, x_0)\big), \tag{23}$$
$$\mathbf{x}_2 = \big((t_0, x_0), (t_*, x_0), (t_1, x_0)\big), \tag{24}$$

where $t_0 < t_* < t_1$. Clearly, as one has an additional observations, the two time series contain different information. However, all interpolation schemes given in Section 3 result in the interpolation $X_{\mathbf{x}}(t) = (t, x_0)$ for $t \in [t_0, t_1]$. As such, the map $\mathbf{x} \to X_{\mathbf{x}}$ is not injective.

### B.2.2 Observational frequencies guarantee uniqueness

First, we define the measurement intensity vector $c_{n,j}$ to be a count of the number of observations of the $j$-th feature up until time $n$. Explicitly we have

$$c_n^j = \sum_{t=0}^{i} \mathbb{1}_{\{x_n^j \neq *\}}. \tag{25}$$

This is often also termed the *measurement intensity vector.*

Suppose that we have two time series, $\mathbf{x} = \{(t_0, x_0, c_0^x), \dots (t_n, x_n, c_n^x)\}$ and $\mathbf{y} = \{(t_0, y_0, c_0^y), \dots (t_n, y_n, c_n^y)\}$, where $c_i^*$ denotes the corresponding measurement intensity vector. Assuming that $\mathbf{x} \neq \mathbf{y}$, then we must have that either

1. There exists some $t_i$ for which $x_i \neq y_i$.

2. There exists at least one $t_i$ for which $\mathbf{x}$ is measured and $\mathbf{y}$ is not (we take $\mathbf{x}$ over $\mathbf{y}$ here wlog).

This proof proceeds in two parts. Firstly, we show injectivity of the map $\mathbf{x} \to X_{\mathbf{x}}$ by showing the interpolations from two different sets of data must be different. Secondly, we prove uniqueness of the map $X_{\mathbf{x}} \to$ Signature$(X_{\mathbf{x}})$ by showing that the schemes from Section 3 cannot contain tree-like pieces (see Hambly & Lyons (2010)).

Let $t_i$ denote the first time for which one of the two conditions above is violated. If 1 holds, then $x_i \neq y_i$ and so $X_{\mathbf{x}}(t_i) \neq X_{\mathbf{y}}(t_i)$ there. Similarly, if 2 holds, then we will have that $c_i^x \neq c_i^y$ there and so again $X_{\mathbf{x}}(t_i) \neq X_{\mathbf{y}}(t_i)$. This then proves that

$$\mathbf{x} \to X_{\mathbf{x}} \text{ is injective with respect to } \mathcal{X}, \tag{26}$$

provided we include observational intensities. Here $\mathcal{X}$ is as defined in Section 2.

To show additionally that we have injectivity of the signature, we need to argue that the path cannot contain tree-like pieces. For a path to contain a tree-like piece, the path must exactly trace itself backwards which is a strong condition Hambly & Lyons (2010). This can only be true if there is a point of turnaround for all dimensions simultaneously, we will now argue why this cannot be the case for any of our proposed control paths if the observational intensities $c_i$ are included.

Recall that $c_i \in \mathbb{R}^v$ increments by one according to the channels of $x_i$ that are measured at $t_i$. Let $c_{i,j} \in \mathbb{R}$ denote the $j^{\text{th}}$ channel of $c_i$. By definition, at every time $t_i$ at least one feature of $c_i$ is updated. Therefore we can find $j_1, \dots, j_n$ with each $j_k \in \{1, \dots, v\}$ such that $c_{i,j_i}$ is incremented.

For linear and cubic Hermite controls, there is always a dimension in $c_i$ that is increasing. This is a sufficient condition for there to be no tree-like pieces. Similarly, for rectilinear, always at least one of the time dimension or a $c_i$ dimension is increasing, and so rectilinear controls too have no tree-like pieces. This condition is unknown however for natural cubic splines.

Together, this shows that provided we include $c_i$ then all of our considered control schemes result in

$$\mathbf{x} \to (x_0, \text{Signature}(X_{\mathbf{x}})) \text{ is injective with respect to } \mathcal{X}, \tag{27}$$

holding for linear, cubic Hermite, and rectilinear controls.

Note that inclusion of $c_i$ is by no means the only way to guarantee this property, however, it represents a sensible approach from which uniqueness can be achieved.

### B.3 Brownian motion toy example

Each Brownian path, $x_t^{(i)}$, was simulated so that it is observed at times $\{0, 0.5, 1\}$. The labels at each time point is whether $x_t^{(i)} > 0$ (in which case the label is 1) or not (in which case the label is 0). Essentially the Neural CDE should learn the Gaussian distribution. Due to the independent Gaussian increments of Brownian motion, an online method can do no better than 75%.

We have a training size of 4096 and a test size of 1024 and the model was trained with the Adam optimiser on batches of 64 across 100 epochs. For the neural network, there are 10 hidden channels, with a one hidden layer of 256 neurons.

The ODE solver used was the Dormand-Prince 5(4) ("`dopri5`") scheme with an absolute tolerance of $10^{-9}$ and a relative tolerance of $10^{-7}$. All simulation and training took place on a computer with two Quadro GP100's.

Code is available at [redacted].

## C  Experimental details

### C.1  Brownian motion toy example

Each Brownian path, $x_t^{(i)}$, was simulated so that it is observed at times $\{0, 0.5, 1\}$. The labels at each time point is whether $x_t^{(i)} > 0$ (in which case the label is 1) or not (in which case the label is 0). Essentially the Neural CDE should learn the Gaussian distribution. Due to the independent Gaussian increments of Brownian motion, an online method can do no better than 75%.

We have a training size of 4096 and a test size of 1024 and the model was trained with the Adam optimiser on batches of 64 across 100 epochs. For the neural network, there are 10 hidden channels, with a one hidden layer of 256 neurons.

The ODE solver used was the Dormand-Prince 5(4) ("`dopri5`") scheme with an absolute tolerance of $10^{-9}$ and a relative tolerance of $10^{-7}$. All simulation and training took place on a computer with two Quadro GP100's.

Code is available at [redacted].

### C.2  General notes for practical datasets

We begin with details common to all experiments.

**Code**   All code is available at `github.com/jambo6/online-neural-cdes`.

**Normalisation**   For all datasets, each channel was normalised to have zero mean and unit variance.

**Data splits**   For all problems we randomly split the dataset into 3 (stratified by label for classification problems) with 75%/15%/15% in training/validation/testing respectively.

**ODE Solvers**   When comparing between interpolation schemes the integral in Equation (1) was solved using the Dormand-Prince 5(4) ("`dopri5`") scheme with an absolute tolerance of $10^{-5}$ and a relative tolerance of $10^{-3}$. For the MIMIC-IV benchmarking results (Table 9) we used the Runge-Kutta-4 ("`rk4`") scheme.

**Optimiser**   The optimiser used for every problem was Adam (Kingma & Ba (2014)).

**Training details**   All models were run with batch size 1024 for up to 1000 epochs. If training loss stagnated for 15 epochs the learning rate was decreased by a factor of 10. If training loss stagnated for 60 epochs then model training was terminated and the model was rolled back to the point of lowest validation loss.

**Computing infrastructure**   All experiments were run on two computers. One was equipped with two Quadro GP100's, the other with one NVIDIA A100-PCIE-40GB.

### C.3   Hyperparameter selection

Hyperparameters for all datasets and models were found using a Bayesian optimisation with 20 trials using the Adaptive Experimentation Platform (ax, 2021) framework.

For Neural CDEs, hyperparameters were optimised separately for the the natural cubic, linear, and rectilinear schemes, with non-natural cubic inheriting the hyperparameters from linear.

We admit a slight error in missing off the learning rate optimisation for the Neural CDE models (it was fixed at 0.005), however if anything, this biases against the Neural CDE results.

The search space for all regular datasets was `hidden_dim in range [32, 256]`, `hidden_hidden_dim in range [32, 192]`, `num_layers in range [1, 4]`, and `learning_rate = 0.005`.

The search space was modified slightly for the MIMIC-IV tasks so as to work within the memory requirements of the GPUs. We used `hidden_dim in range [32, 128]`, `hidden_hidden_dim in range [32, 128]`, `num_layers in range [1, 4]`, and `learning_rate = 0.005`. The learning rate was dropped by a factor of 10 for training the actual to account for observed overfit.

For the GRU variants (GRU, GRU-dt, GRU-dt-intensity, GRU-D) we used `hidden_dim in range [32, 512]`, `learning_rate in range [0.0001, 0.1]`.

For the ODE-RNN we used `hidden_dim in range [32, 256]`, `hidden_hidden_dim in range [32, 196]`, `num_layers in range [1, 4]`, `learning_rate in range [0.0001, 0.01]`.

Final values for all hyperparameters are given in Tables 10 and 11

### C.4   MIMIC-IV

The Medical Information Mart for Intensive Care (MIMIC) IV dataset comprises of de-identified patient-level electronic health record (EHR) data from over 50,000 patients that stayed in ICUs at the Beth Israel Deaconess Medical Center, Boston, Massachusetts between 2008 and 2019. MIMIC-IV builds on its predecessor MIMIC-III. Improvements include approximate year of admission has been made available (previously obfuscated as pat of de-identification), and there are more granular information on intake of medications through use of new tables.

Specifically, MIMIC-IV contains information on 53,150 patients, across 69,211 hospital admissions and 76,540 ICU stays. These are stored in relational tables in SQL format. Our first stage of data processing involves merging the information and pivoting the data such that we have time series data with each channel/dimension a separate variable of interest. We extracted data on vital signs, laboratory reading, ventilation devices. The full list of extracted features can be seen at `github.com/jambo6/online-neural-cdes/main/get_data/mimic-iv/features.py`.

We use the MIMIC-IV dataset on three problems of interest, namely mortality prediction, length of stay (LOS) prediction and sepsis prediction.

For all tasks we removed patients whose stay length was greater than 72 hours (as the data has a very long tail in this regard), and additionally required a stay length greater than 4 hours and measurements at at least 4 different times. As the problems are different, the exclusion criteria are slightly different in each case. The exclusion concept diagram can be seen in Figure 3

| Dataset | Model | Interpolation | Hidden | Hidden hidden | Num layers | LR |
|---|---|---|---|---|---|---|
| BeijingPM10 | NCDE | Natural cubic | 256 | 168 | 1 | 0.0005 |
| | NCDE | Linear | 252 | 120 | 1 | 0.0005 |
| | NCDE | Rectilinear | 249 | 170 | 1 | 0.0005 |
| BeijingPM2pt5 | NCDE | Natural cubic | 180 | 32 | 3 | 0.0005 |
| | NCDE | Linear | 256 | 196 | 1 | 0.0005 |
| | NCDE | Rectilinear | 256 | 171 | 1 | 0.0005 |
| CharTraj | NCDE | Natural cubic | 238 | 194 | 1 | 0.0005 |
| | NCDE | Linear | 249 | 170 | 1 | 0.0005 |
| | NCDE | Rectilinear | 148 | 89 | 2 | 0.0005 |
| LOS | NCDE | Natural cubic | 108 | 40 | 4 | 0.0005 |
| | NCDE | Linear | 65 | 119 | 2 | 0.0005 |
| | NCDE | Rectilinear | 32 | 126 | 4 | 0.0005 |
| | ODE-RNN | – | 32 | 168 | 1 | 0.002268 |
| Mortality | NCDE | Natural cubic | 41 | 55 | 4 | 0.0005 |
| | NCDE | Linear | 33 | 91 | 2 | 0.0005 |
| | NCDE | Rectilinear | 70 | 55 | 2 | 0.0005 |
| | ODE-RNN | – | 32 | 123 | 4 | 0.1 |
| Sepsis | NCDE | Natural cubic | 82 | 65 | 2 | 0.0005 |
| | NCDE | Linear | 82 | 65 | 2 | 0.0005 |
| | NCDE | Rectilinear | 82 | 65 | 2 | 0.0005 |
| | ODE-RNN | – | 215 | 98 | 2 | 0.001083 |
| SpeechCommands | NCDE | Natural cubic | 56 | 125 | 3 | 0.0005 |
| | NCDE | Linear | 77 | 118 | 3 | 0.0005 |
| | NCDE | Rectilinear | 113 | 124 | 3 | 0.0005 |

Table 10: Final hyperparameter values for the ODE models (Neural CDE and ODE-RNN).

| Dataset | Model | Hidden dim | Learning rate |
|---|---|---|---|
| LOS | GRU | 435 | 0.017858 |
| LOS | GRU-dt | 299 | 0.001359 |
| LOS | GRU-dt-intensity | 294 | 0.000753 |
| LOS | GRU-D | 408 | 0.000143 |
| Mortality | GRU | 432 | 0.025304 |
| Mortality | GRU-dt | 508 | 0.000158 |
| Mortality | GRU-dt-intensity | 196 | 0.017743 |
| Mortality | GRU-D | 398 | 0.000161 |
| Sepsis | GRU | 362 | 0.000697 |
| Sepsis | GRU-dt | 362 | 0.000697 |
| Sepsis | GRU-dt-intensity | 362 | 0.000697 |
| Sepsis | GRU-D | 362 | 0.000697 |

Table 11: Final hyperparameter values for the GRU variants.

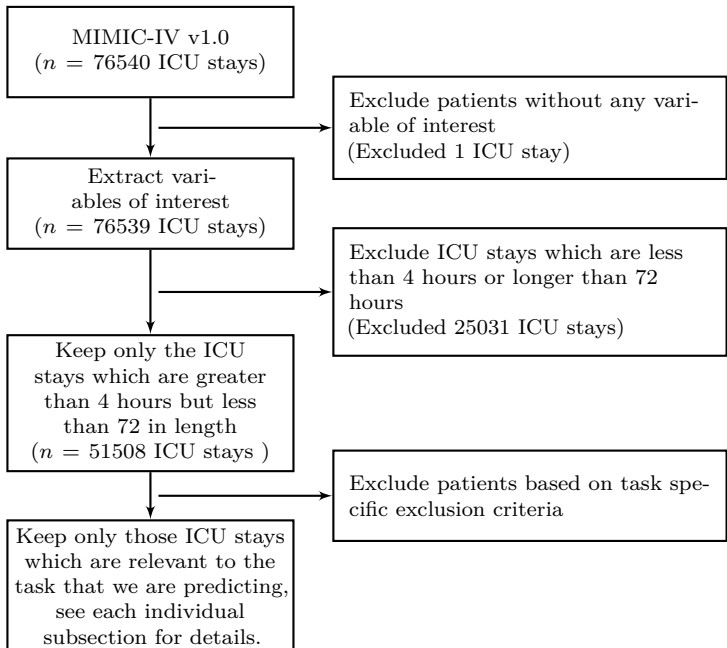

Figure 3: Study flow diagram

**Mortality**   Here we attempted to predict in hospital mortality for each patient. We required measurements at at least 4 unique times, and at least 4 hours worth of data. This resulted in 50,538 ICU stays.

**LOS**   We predicted length of stay given the first 24 hours of data for patients who had a length of stay between 24 and 72 hours. This resulted in 16,054 ICU stays.

**Sepsis**   Historically, there have been many standards used to define sepsis from the data as a way to operationalize research. One of the latest and commonly used definition is Sepsis-3 (Singer et al., 2016) which describes sepsis as "life-threatening organ dysfunction caused by a dysregulated host response to infection." For the purpose of operationalization, organ dysfunction is identified by an increase in Sequential Organ Failure Assessment (SOFA) score (Vincent et al., 1996) by 2 points or more consequent to the infection.

As the focus in (Singer et al., 2016) was not the detection of onset, the onset time was not made explicit. As a result, there have been various interpretations, for example at the point of SOFA score increase, or the earlier of time of suspected infection and time SOFA score increase in (Desautels et al., 2016) and (S et al., 2018) respectively. From a clinical perspective, it makes sense to define the onset of sepsis as the time when we see organ dysfunction (as indicated by SOFA score) provided that we see evidence of suspicions of infection around the same time (thorough the use of antibiotics and cultures). Therefore this is the way we have chosen to define the time of onset of sepsis, in line with Desautels et al. (2016).

To determine the time of sepsis onset, we looked for a suspicion of infection and a deterioration in the SOFA score. To identify a suspicion of infection, we first find the times that lab cultures were taken. We also find the start of new antibiotic treatments, and require that there are two doses of antibiotics given to the patient within a 96 hour window, otherwise that time is not used in suspicion of infection. If an antibiotic treatment has commenced within 24 hours before or 72 hours after taking a culture sample, then the earlier of the two times defines a time of suspected infection. We then look at a patient's SOFA scores and see if there is an increase within 24 hours before or 48 hours after the time of suspected infection. We note that the choice of these time windows are aligned with Singer et al. (2016) but other choices are used in the literature. The time of sepsis onset is then given by the earlier time, i.e. min(time of suspected infection, time of SOFA deterioration).

| Control | AUC | NFEs$\times 10^3$ |
|---|---|---|
| Natural cubic | $0.779 \pm 0.003$ | $\mathbf{16.2 \pm 0.1}$ |
| Linear | $\mathbf{0.784 \pm 0.002}$ | $27.8 \pm 0.3$ |
| Cubic Hermite | $0.782 \pm 0.002$ | $20.2 \pm 0.4$ |
| Rectilinear | $0.772 \pm 0.004$ | $136.7 \pm 4.5$ |

Table 12: Comparison of the control signals on the sepsis dataset. This complements Tables 7 and 8 and is included in the appendix for completeness since it does not alter the narrative of the paper.

The older, more obsolete EHR data from the CareVue system (2003-2008) has been removed in MIMIC-IV. This has meant that more patients can be included in the sepsis analysis as previously the CareVue data lacked the necessary antibiotic information required for the Sepsis-3 definition.

As well as using a cut off at 72 hours, we also removed patients who were prescribed antibiotics before their ICU admission time and those developed sepsis within 4 hours of their ICU stay. This resulted in 45,218 ICU stays.

### C.5 Additional results

We give the interpolation results for the sepsis task in Appendix C.4. This is left out of the main paper only for reasons of space, but we see that it tells much the same story as the Mortality and LOS datasets. One note is that the rectilinear interpolation batch size had to be reduced from 1024 to 512 as it would not fit within GPU memory.

