# OpenReview forum: "On the Choice of Interpolation Scheme for Neural CDEs"
_TMLR — Accepted by TMLR_

### Review · Reviewer_T4bA · 2022-07-05

**Summary Of Contributions:**

- The paper analyses the choice of the interpolation schemes that power Neural Controlled Differential Equations (NCDEs). To that end, the paper proposes four theoretical properties that control paths should ideally satisfy: adapted measurability, smoothness, boundedness, and uniqueness.
- Of these four, the paper is particularly focused on the first property, which (informally) requires that the NCDE model is able to make online predictions. Existent interpolation schemes (e.g. cubic splines) are known to violate this property and, therefore, existent Neural CDEs cannot be used for online prediction tasks. This can also be interpreted as existent NCDEs being implicitly non-causal (outputs at time $t$ rely on the value of the time-series at a later time $t' > t$).
- The paper proposes two new interpolation schemes, cubic Hermite splines with backward difference and rectilinear control, which satisfy the four properties (to various degrees). Most importantly, these schemes can be used for different types of online tasks.
- The experimental analysis shows improved performance on medical time-series data and other benchmarks.
- Finally, the paper proposes a set of guidelines for practitioners explaining which interpolation scheme should be chosen for a given task.

**Requested Changes:**

All the points in the weaknesses section above are easily addressable and I believe would strengthen the paper. They are not however critical for acceptance.

**Strengths And Weaknesses:**

# Strengths

- The paper addresses an important problem that would be of interest to anyone working with time series: making Neural CDE models applicable to online prediction tasks.
- The four theoretical properties for control paths that the paper introduces offer a nice framework for anchoring the discussion in the entire paper. Furthermore, the paper does a good job of justifying these properties theoretically, which could also be helpful for discussing and comparing Neural CDE models in the future.
- The paper proposes two new simple and effective interpolation schemes that satisfy the discretely online and, respectively, continuously online properties. While the cubic Hermite splines with BD are discretely online, unlike linear control, it is smooth, which leads to NFE improvements. At the same time, rectilinear control is the first interpolation scheme for Neural CDEs that is fully online (i.e. continuously online).
- The paper contains many empirical experiments showing the benefits of the proposed schemes in terms of the prediction error and the number of function evaluations compared to other interpolation schemes.
- Figures one and two are very good at explaining the intuition behind the introduced concepts.
- The paper is well written and strikes the right balance between mathematical rigour and readability.

#  Weaknesses

- The presentation of the rectilinear control method can be significantly improved to better explain the intuition behind what is going on. This is partially achieved via the last subfigure of Figure 2 which uses the notion of "virtual time". I think this is quite a nice way to understand what the method is doing, however, what is meant by "virtual time" is not explained textually anywhere and the term appears only in the figure.
- The paper does not explain how the dataset is split more precisely. It is only mentioned that "70%/15%/15%" splits are performed. Are these random splits for the points in each time series in the dataset or is the splitting performed in a particular way? I also checked the appendix, but I could not find more details there either.
- Something that could have been useful to see is time-series extrapolation experiments (even synthetic ones). I think the newly proposed interpolation schemes are more suitable for time-series extrapolation than existent NeuralCDEs, which can only make predictions up to the time of the maximum available time-point.

---

> ### Author Response · Authors · 2022-08-01
> **Response to reviewer T4bA**
>
> We have updated our description of virtual time in the section on “rectilinear control”. I think the main cause of confusion is that we erroneously termed the same concept “integration time” in the body of the text, but called it “virtual time” in the figure caption. This has now been remedied and we now only refer to virtual time. Thank you for pointing this out.
>
> The 70/15/15 split is random with label stratification for classification tasks, so the same proportion of positive/negatives are in each split (approximately). This is stated in appendix C.2 “Data splits”.
>
> We certainly agree that time series forecasting represents a good use case for these models. However, we feel demonstration that our "everywhere online" model achieves performance as-good-as or exceeding existing non-online benchmarks is sufficient for this paper.

---

> > ### Comment · Reviewer_T4bA · 2022-08-08
> > **Response**
> >
> > Thank you for addressing my comments!

---

### Review · Reviewer_cpiv · 2022-07-07

**Summary Of Contributions:**

This paper investigates different strategies for allowing online sequential processing of data with neural CDEs. The authors first show why the classical Neural CDE implementation fails to deal with online streams of data due to the global temporal dependencies of cubic splines. They then go on with providing desirable properties for an online interpolation strategy.

The authors then propose 3 strategies : linear, cubic hermite and rectilinear.

They evaluate the strategies on regression and classification tasks both from synthetic and real-world data.

**Broader Impact Concerns:**

I have no broader impact concern.

**Requested Changes:**

I've described changes i'd like to see above but in summary :

- Elucidating why the natural splines performs worse than the continuous interpolation methods.
- Being more explicit about the relevance of a continuous online setup for classification and regression.
- Impact of interpolation strategy from a computational perspective.

What is more, in Figure 1, I'm also not convinced that future values of the control $X$ will impact the current value of $z(t)$ as the ODE is by definition causal. I'd like the author to either explain me why this is the case or to adapt the figure accordingly.

**Strengths And Weaknesses:**

Strength.

The inability to process online streams of data is an important limitation for Neural CDE and the motivation for this work is clear. I appreciate the rigor in defining the criteria that the interpolation strategies should statisfy. The experimental section is solid, with various data sets and tasks and emphasizing the computational speed trade-offs. The paper is also well written.

Weaknesses.

I understand that natural splines would get better performance to the leakage of future information in the stochastic process experiment. However, I don't understand why this effect is not present in the other experiments. Indeed, in the other tasks, Neural splines consistently underperforms the online interpolation strategies, which is counter-intuitive. I would recommend to expand on this in the main text or explaining the specificities of the experimental setup that leads to this effect.

To me, there is little motivation for the continuous online setup in irregular classification or regression exeperiments. Indeed, in between observations, you don't really care about the part of the time series  between now and the last observation because there is just not information there. The only case where I would imagine a difference would be in forecasting (which you don't address here). I think this is worthwhile discussing this in the paper as well, as it puts the relevance of the rectilinear approach into perspective.

It seems that in your experiments, you are computing the hermite splines a priori (which makes sense). However, in a real continuous setup, we would have to compute this on the fly. Could you maybe elaborate on the extra computational complexity that this entails, if any ?

---

> ### Author Response · Authors · 2022-08-01
> **Response to reviewer cpiv**
>
> Regarding the performance of natural cubic splines being worse on other experiments: very good question. The BM simulation is the only experiment we do which makes predictions along multiple timepoints. We do this to illustrate the data leakage problem. The rest of the experiments in the paper were all final time prediction, we have defined the experimental set-up in this way as we wanted to compare the performance of the different interpolation schemes on an 'equal footing'.
>
> A few areas where continuously online solutions would be of importance:
> 1. Anywhere execution is not necessarily optimal at datapoints. A couple of examples:
>     - Optimal time to administer a drug or treatment.
>     - Optimal time to execute a financial trade.
> 2. Incorporation of uncertainty. Whilst not considered here, a simple method would be to track an estimate of the prediction standard deviation which would increase between measurements.
>
> We do not see a fundamental difference between classification and regression in this scenario. Take the prediction of sepsis for example. It's possible we might want a continuous risk estimate changing in time (regression), but it is also possible we might want a binary prediction about whether drug X should be administered at the current time. The latter would constitute a classification task, but it is one that is "continuously online". As such, we think distinguishing between the ideas of continuously online, discretely online, and offline is more appropriate.
>
> In the experiments, we are indeed constructing the interpolations upfront. Cubic Hermite is just a function of the last point and the left derivative, along with the location of the next point. Whilst it is slightly more computationally expensive than updating linearly, it is still O(d) where d is the dimension of the signal. We have added a short note in the interpolation section about this.
>
> With regards to your comment about Figure 1 and causality of ODEs, we see by equation (1) that $z(t)$ depends on $\frac{d X_x}{ds}$. To be precise, the latent space is dependent on the gradient of the interpolated control path, which in an offline scheme will depend on future values (in order to match various gradients).

---

### Review · Reviewer_TQQf · 2022-07-20

**Summary Of Contributions:**

This paper considers interpolation schemes for neural CDEs, defined as a sequence regression model where the hidden state is a dynamical system of the form $z(t) = z(0) + \int f_\theta (z(s)) dX_x(s)$. The last term being a suitable interpolation of the observed discrete time-series data $x$. This paper studies the choice of interpolation schemes that forms $X_x$. A primary issue under consideration is the application to online prediction, where continuous predictions need to be generated without looking at future discrete observation points. The authors defines these requirements as 1) continuous online (the entire prediction of $z(t)$ only depends on observations of $t_i < t$) or the weaker 2) discretely online (the prediction of $z(t_j)$ only depends on observations $t_i < t_j$). The authors show that the original choice of natural cubic splines do not satisfy these requirements. Thus, the authors propose two other types of interpolation schemes: 1) cubic hermite splines with backward differences (splines coefficients only depend on the neighboring knots, and additional degrees of freedom fixed by matching the slopes of linear interpolation) and 2) rectilinear interpolation (piece-wise constant interpolation). They demonstrate that these choices indeed satisfies the discretely (resp. continuously) online properties, and have favorable performance with respect to baselines such as natural cubic splines interpolation, as well as non-CDE benchmarks.

**Broader Impact Concerns:**

None identified.

**Requested Changes:**

Below, I have some questions for the authors. If they are addressed adequately, I am happy to recommend acceptance.

1. Eq 1. Would it not be more precise to write the integral as $\int f_{\theta_1} dX_x(s)$ instead of including the derivative $dX_x(s)/ds$. The interpolation schemes considered subsequently in this paper include non-differentiable (pw linear) and discontinuous ones (rectilinear).
2. Integration time: it is mentioned several times that "integration time" varies when different interpolation schemes are used. Can the authors make this statement precise, since different integrators in general will have different computational time, and their dependence on properties such as smoothness/gradient norms are all different. For example, the computation time of RK4 with or without adaptive step size selection will vary differently with respect to changing signal smoothness. Moreover, since the model has to be trained, wouldn't the training time (affected by efficiency of gradient evalutions with respect to $\theta_1,\theta_2$) also play a role?
3. Observational frequencies $c_i$ definition "counts the number of times the channels in $x_i$ have been observed up to $t_i$." By channels, do the authors mean the components of the vector $x_i$? By "number of times" do the authors mean the number of $i$ for which $x_{i,j}$ (jth component) is not $*$? I suggest the authors be a little more precise with the definition (say with a formula), since it is an important object in this study
4. Rectilinear control: how is the proposed interpolation scheme different from a simple piece-wise constant (and left continuous) interpolation (with missing data forward-filled)? Why is there a need to do a linear interpolation holding $t$ constant? It does not appear to affect the integrator how you interpolate in this region. The authors may consider adding more explanation here.
5. Natural extrapolation schemes: while only the rectilinear interpolation is the continuously online choice considered in this paper, one may wonder if a simple natural extrapolation scheme (say matching up to a number of derivatives) may produce different results. This would also be continuously online. Have the authors tested these types of schemes? I suppose they also have the discountinuity issues, and hence slow down integration?
6. Table 6 and Tables 7-8 (NFE) column did not boldface the best result. Is this intentional?


**Strengths And Weaknesses:**

Overall, this paper is well written and the presentation of the results are generally clear. The subject matter is important to practictioners who wish to develop a continuous-in-time prediction model for time-series analysis tasks that cannot look at future observational values. This is a common scenario. The results are well substantiated with different experimental settings. In particular, the paper gives some guidelines on how to choose the appropriate interpolation scheme depending on the requirements of the problem.

The main shortcoming is the lack of theoretical understanding of why different interpolations give different performance, but given the variety of factors that may come into play (choice of integrator, properties of dataset, choice of learning/optimization algorithm, choice of neural network architectures etc.), this is probably a difficult task.

---

> ### Author Response · Authors · 2022-08-01
> **Response to reviewer TQQf**
>
> 1. We may use the second integral form when the control is piecewise continuous (which we always have here) provided we align our integration endpoints with the points of discontinuity (which we always do). We have noted this in "2.2 smoothness", but are happy to expand on this if you think it needs it.
> The suggested form is more general; however, in our experience, we find it tends to confuse people and so we opt for the second version since it covers our controls under consideration.
>
> 2. Our use of the term integration time was somewhat erroneous (see the response to reviewer T4bA). By integration time, we specifically refer to the parts of the parameterisation in the rectilinear interpolation where real (observation) time is held fixed but the features are changed. This was also referred to as "virtual time" in the initial version, and we have now updated so that all instances of integration time are changed to virtual time, which hopefully makes things clearer.
>
> 3. You are correct on both points. We have updated Appendix B.2.2 to provide this information alongside a formula.
>
> 4. The proposed Càdlàg scheme results in a jump at the locations where data arrive. The rectilinear interpolation resolves this by connecting' the discontinuities. This is achieved by introducing a virtual time as an alternative parameterisation that switches between interpolating real (observation) time and interpolating between observation values. Ensuring continuity is the purpose of introducing the virtual time reparameterisation.
>
> 5. Could you explain what you mean by a "natural extrapolation scheme"? I'm not sure how you could define a scheme to match gradients without looking into the future to at least see the next point.
>
> 6. No not intentional. This has been amended, thank you.

---

> > ### Comment · Reviewer_TQQf · 2022-08-04
> > **On extrapolation**
> >
> > Thank you for addressing my comments. By natural extrapolation I mean just using the one-sided (left) derivatives up to a certain order to extrapolate. I suppose this will still introduce discontinuity issues.

---

### Author Response · Authors · 2022-08-01
**Thank you to the reviewers**

We would firstly like to thank each of the reviewers for their time and feedback. Individual responses have been addressed below.

---

### Author Response · Authors · 2022-09-05
**Reupload and modification locations**

We have uploaded a new version with the new updates.

You can see this version here with orange star-marked locations to denote where amendments have been made. These are on pages 6, 8, 9, 16, 17.

https://ncde-tmlr-modification-locations.tiiny.site/

Please let us know if anything further is required.

Thank you again to the reviewers and AC for their time, and for providing a useful and enjoyable discussion period.

---

### Decision · Action_Editors · 2022-09-01

**Recommendation:** Accept with minor revision

**Comment:**

All reviewers appreciated the importance of the topic to practitioners, as well as the experiments and guidelines discussed. The reviewers noted several areas that needed clarification or further discussion.  After the response and discussion, all reviewers were satisfied and recommended acceptance. The authors should prepare a revision of the paper according to the review and response.